# Prolonged Cold Exposure Negatively Impacts Atlantic Salmon (*Salmo salar*) Liver Metabolism and Function

**DOI:** 10.3390/biology13070494

**Published:** 2024-07-03

**Authors:** Isis Rojas, Albert Caballero-Solares, Émile Vadboncoeur, Rebeccah M. Sandrelli, Jennifer R. Hall, Kathy A. Clow, Christopher C. Parrish, Matthew L. Rise, Andrew K. Swanson, Anthony K. Gamperl

**Affiliations:** 1Department of Ocean Sciences, Memorial University of Newfoundland and Labrador, St. John’s, NL A1C 5S7, Canadakgamperl@mun.ca (A.K.G.); 2Aquatic Research Cluster, CREAIT Network, Ocean Sciences Centre, Memorial University of Newfoundland and Labrador, St. John’s, NL A1C 5S7, Canada; 3Cooke Aquaculture Inc., Saint John, NB E2L 3H3, Canada

**Keywords:** winter mortality, winter syndrome, lipid metabolism, fatty liver disease, salmon aquaculture

## Abstract

**Simple Summary:**

Winter-associated mortalities at Atlantic salmon marine aquaculture sites have been reported in the last decade in Atlantic Canada and Iceland. Some fish species develop a condition known as ‘Winter Syndrome’ or ‘Winter Disease’ (WS/WD) when held at cold temperatures that is associated with enlarged and pale livers (fatty liver disease; FLD). To determine if salmon can develop FLD, we measured liver fat content and composition, and the expression of 34 molecular biomarkers of FLD in fish held long-term at 3 °C. Total fat and triacylglycerol levels were 50% higher in fish that showed symptoms of FLD, compared with healthy fish. Also, the expression level of 32 of the 34 selected biomarkers changed in fish with symptoms of FLD. Our results show that prolonged exposure to cold temperatures can lead to the development of FLD in some Atlantic salmon, and suggest that the development of new diets may improve the health and welfare of salmon during the winter.

**Abstract:**

Large-scale mortality events have occurred during the winter in Atlantic salmon sea cages in Eastern Canada and Iceland. Thus, in salmon held at 3 °C that were apparently healthy (i.e., asymptomatic) and that had ‘early’ and ‘advanced’ symptoms of ‘winter syndrome’/’winter disease’ (WS/WD), we measured hepatic lipid classes and fatty acid levels, and the transcript expression of 34 molecular markers of fatty liver disease (FLD; a clinical sign of WS/WD). In addition, we correlated our results with previously reported characteristics associated with this disease’s progression in these same individuals. Total lipid and triacylglycerol (TAG) levels increased by ~50%, and the expression of 32 of the 34 genes was dysregulated, in fish with symptoms of FLD. TAG was positively correlated with markers of inflammation (*5loxa*, *saa5*), hepatosomatic index (HSI), and plasma aspartate aminotransferase levels, but negatively correlated with genes related to lipid metabolism (*elovl5b*, *fabp3a*, *cd36c*), oxidative stress (*catc*), and growth (*igf1*). Multivariate analyses clearly showed that the three groups of fish were different, and that *saa5* was the largest contributor to differences. Our results provide a number of biomarkers for FLD in salmon, and very strong evidence that prolonged cold exposure can trigger FLD in this ecologically and economically important species.

## 1. Introduction

Atlantic salmon (*Salmo salar*) aquaculture is an important industry with global production and market values of approximately 2.8 million tonnes and USD 20 billion in 2021, respectively [1]. Large-scale production of this species occurs in several regions of the world, and thus, salmon are exposed to varied temperature ranges in the coastal marine locations where the grow-out phase of culture typically takes place. Given concerns about ocean warming and the impacts of high water temperatures on the health, welfare, and survival of farmed Atlantic salmon (e.g., see [2]), a number of studies have: examined the effects of elevated water temperatures on this species’ production characteristics (i.e., growth, food consumption, feed conversion efficiency), physiology, and immune function; determined its acute and chronic upper thermal tolerances [3,4,5,6,7,8,9]; and investigated whether growth and survival at high temperatures could be selected for in-breeding programs [7,10]. However, winter-related losses in Eastern Canada and Iceland (where the industry is expanding) in recent years [11,12,13,14,15,16] indicate that challenges to rearing salmon at cold temperatures must also be addressed to improve salmon welfare and the industry’s sustainability and profitability. Low temperatures were identified as a challenge to salmonid aquaculture several decades ago [17], and there has been considerable research on several aspects of ‘cold stress’ and ‘cold shock’ in fishes [18,19]. However, there has been surprisingly little effort to understand the temperature(s) below which sub-lethal effects occur in Atlantic salmon, their severity, and the underlying causes of mortality during the winter at cage sites.

Recently, Vadboncouer et al. [20,21,22] conducted several studies to examine the effects of short-term and long-term exposure to cold temperatures on Atlantic salmon production and physiology, and reported that prolonged exposure to low temperatures can have a number of negative consequences. These authors first exposed salmon post-smolts to a gradual decrease in temperature of 1 °C·week^−1^ from 8 °C that mimicked fall/winter temperature changes at Newfoundland cage sites [23]. The salmon began to reduce feed intake at 6 °C and stopped feeding by ~2 °C. At 4–5 °C, cellular stress (as evidenced by increased liver *hsp70* and *hsp90* transcript expression) and an osmoregulatory disturbance were first noted, and these became progressively worse as the fish were cooled further. Finally, plasma cortisol levels were elevated at 2 and 1 °C, and at 1 °C the fish were found to have enlarged livers (i.e., an increased hepatosomatic index, HSI) and what appeared to be the beginning of an opportunistic infection characterized by fin rot and skin erosion/ulceration of the head. In addition, Vadboncouer et al. [20,21,22] sampled salmon from a population of fish that had been held at a temperature of 3 °C for >5 weeks and was experiencing significant mortalities (~30%); the sampled fish classified as healthy (asymptomatic) or exhibiting ‘early’ (only lethargy/erratic swimming) or ‘advanced’ (including the above signs of an opportunistic infection) symptoms. Fish with ‘early’ and ‘advanced’ symptoms had livers that were larger (HSI ~2.5 vs. 1.6 in asymptomatic fish), pale, and friable. Further, analysis of plasma samples showed that symptomatic fish were suffering from an ionoregulatory disturbance, and had high plasma levels of cortisol (~100 ng·mL^−1^) and aspartate aminotransferase (AST, a key biomarker of liver damage).

The above symptoms are characteristic of a pathological condition termed ‘Winter Syndrome’ or ‘Winter Disease’ (WS/WD) that has been described in cultured gilthead sea bream (*Sparus aurata*) for approximately 2 decades [24,25,26], and has also recently been identified in yellow drum (*Nibea albiflora*) [27,28]. Further, Dessen et al. reported that changing Atlantic salmon to a diet with a higher fat content in winter (at ~6 °C) resulted in the mortality of seemingly healthy fish, and that moribund fish had higher values for HSI, liver fat content, and plasma AST levels [29]. Based on the above data, it appears that Atlantic salmon are susceptible to fatty liver disease (FLD) and WS/WD when exposed to cold temperatures in culture.

As part of their study, Vadboncoeur et al. [22] sampled liver from fish classified as ‘healthy’ and those exhibiting ‘early’ and ‘advanced’ symptoms. Using these samples for lipid composition and transcript expression analyses, we addressed two questions in this study. First, was the increase in HSI in symptomatic fish related to changes in liver lipid classes and fatty acid (FA) profile? Second, were differences in the transcript expression levels of key hepatic genes consistent with what has been described for other fish species that developed WS/WD, and in mammalian models of NAFLD (non-alcoholic fatty liver disease)? Further, we used multivariate statistics to link previously collected phenotypic data (Vadboncoeur et al. [22]) with the newly generated lipid composition and gene expression results to understand the progression of FLD in this species. Collectively, these data: (1) show that high liver TAG content is associated with increased inflammation and fat accumulation in salmon at cold temperatures, and that high HSI and AST values are linked to an inflammatory response and low antioxidant capacity in this organ; and (2) furnish compelling evidence that FLD and WS/WD can afflict Atlantic salmon at cold temperatures. In addition, they provide novel insights into the mechanisms associated with the poor health and compromised metabolic function of fishes at temperatures near their lower thermal limit, and point to potential strategies (incl. the use of functional feed ingredients) that the salmon aquaculture industry might use to reduce the incidence of this debilitating fish health/welfare challenge.

## 2. Materials and Methods

### 2.1. Animals

Atlantic salmon used for this study were obtained from the same population as detailed previously [22]. The Atlantic salmon were raised in a commercial hatchery in New Brunswick (Canada), and subsequently shipped as pre-smolts to the Dr. Joe Brown Aquatic Research Building (JBARB) at the Ocean Sciences Centre (Memorial University of Newfoundland and Labrador). At the JBARB, the fish were initially maintained in a 3000 L tank provided with flow-through seawater (32 ppm salinity; >95% air saturation) and a 12 h light:12 h dark photoperiod. During this period, the salmon were fed twice a day, until apparent satiation, with 3 mm pellets (Signature Salmon, Northeast Nutrition, Truro, NS, Canada). The specific formulation is provided in [22].

### 2.2. Experimental Protocol and Sampling

Two hundred and twenty-five of these salmon (175 ± 24 g, mean weight ± standard error) were evenly distributed into three 1000 L tanks (75 fish per tank), supplied with 7.5 L·min^−1^ of seawater (~6 °C; 32 ppm salinity) at ~100% air saturation, and a 12 h light:12 h dark photoperiod. The salmon were acclimated to these tanks for three weeks, then the water temperature was gradually decreased (by 0.3 °C per day) until it reached 3 °C, and maintained at this temperature. Temperature and oxygen levels in the tanks were recorded daily using a hand-held meter (YSI, ProODO, OH, USA), and the fish were fed to apparent satiation every second day with the same salmon diet as above.

After five weeks at 3 °C, mortality spiked suddenly, with a number of fish showing signs of ill health (first lethargy and erratic swimming, then ulceration of the skin and head, and fin rot; see [22]). At this point, 27 fish were euthanized in 0.4 g·L^−1^ MS-222 (Syncaine, Syndel Laboratories Ltd., Vancouver, BC, Canada), sampled for blood (by caudal puncture) and liver, and separated into three groups (n = 9 fish per group). The first group of fish were apparently healthy, and were designated as being ‘asymptomatic’, and served as control fish. The asymptomatic fish had no abnormal physical or behavioral symptoms. The symptomatic fish had symptoms that ranged in severity; from those that were only lethargic and swimming at the water’s surface, to ones with extensive head erosion/ulceration. Consequently, the symptomatic fish were divided into two groups using a scoring scheme adapted from [30]. All salmon were examined for three characteristics: snout/head damage, fin damage, and skin lesions. Each characteristic was scored from 0 to 3, with 0 being an absence of damage, and 1 to 3 going from least to most severe. Further, if fin damage was active, as opposed to healed, the fin damage score was doubled. The three scores for each fish were summed to give a total individual score. The 9 fish with the lowest scores were placed in the ‘early’ symptom group (average score 3.8 ± 0.7), and the 9 fish with the highest scores (average score 9.0 ± 0.7) were put in the ‘advanced’ symptom group.

Blood was obtained by caudal puncture within <3 min of the fish being netted using syringes containing lithium heparin and 21-gauge needles. The fish were then measured for weight, fork length and liver weight. Finally, the liver was removed using surgical instruments cleaned with RNase AWAY^®^ (Sigma Chemical Co., Oakville, ON, Canada). Approximately 100 mg samples were placed into 1.5 mL RNase-free tubes, quickly frozen in liquid nitrogen, and then stored at −80 °C until RNA preparation. The remaining liver was flash-frozen for the analysis of lipid classes and FAs.

### 2.3. Lipid Class and Fatty Acid Analyses

Lipid extraction and FA derivatization were performed as described previously [31]. Briefly, all utensils were lipid-cleaned (three times with methanol followed by three times with chloroform or heated in a muffle furnace to 550 °C for 6 h), and the samples were kept on ice during the homogenization process and were blanketed with nitrogen gas after all steps. Chloroform:methanol:water (8:4:3) mixtures were used for the extraction process, and Hilditch’s reagent (1.5 H_2_SO_4_: 98.5 anhydrous MeOH) was used to derivatize FAs in 50 µL of the lipid extract for 1 h at 100 °C. Transesterified samples were analyzed using a HP 6890 gas chromatograph (Agilent Teachnologies, Santa Clara, CA, US) and a Zebron ZB-WAX plus™ (30 m × 0.32 mm × 0.25 µm) column (Zebron, Phenomenex, Aschaffenburg, Germany). Derivatized samples were injected at 65 °C, and the temperature was increased at a rate of 40 °C·min^−1^ to 195 °C, and then increased to 220 °C at a rate of 2 °C·min^−1^. The hydrogen carrier gas flow was 2 mL·min^−1^, and the starting temperature of the injector was 150 °C with an increase of 120 °C·min^−1^ to 250 °C. The detector temperature was kept at 260 °C. The peaks obtained were compared to those obtained using standards from Supelco (Bellefonte, PA, USA): 37 component FA methyl ester (FAME) mix (Product number 47885-U), PUFA 3 (product number 47085-U), and PUFA 1 (product number 47033-U). Chromatograms were integrated using Chromatography Data Systems Open Laboratory CDS, and the FA data were calculated as an area percent of FAME.

### 2.4. Analysis of mRNA Expression of Winter Syndrome-Related Genes

#### 2.4.1. RNA Preparation

Each liver sample was placed in a 2 mL RNase-free Eppendorf^®^ tube containing 400 µL of TRIzol™ Reagent (Invitrogen/Thermo Fisher Scientific, Burlington, ON, Canada) and a 5 mm stainless steel bead (QIAGEN, Mississauga, ON, Canada), and then homogenized at 25 Hz for 2.5 min using a TissueLyser II (QIAGEN). An additional 400 µL of TRIzol™ was added, the tissue was homogenized once more under the same conditions, and then stored at −80 °C. Each sample was subsequently thawed on wet ice and further homogenized through a QIAshredder (QIAGEN) spin column. Two hundred µL of TRIzol™ was then added, and RNA extraction was performed following manufacturer’s instructions. The total RNA pellet was then resuspended in nuclease-free UltraPure™ distilled water (Invitrogen/Thermo Fisher Scientific).

An additional step was performed to purify the samples (modified from [32]) as the liver is a lipid-rich tissue and these components interfere with column purification. Briefly, 150 µg of total RNA was diluted in water up to a final volume of 300 µL in a 1.5 mL RNase-free Eppendorf^®^ tube. An equal volume (300 µL) of phenol:chloroform:isoamyl alcohol (125:124:1, pH 4.7) (AM-9720, Ambion/Thermo Fisher Scientific) was then added, the tubes vortexed for 30 s, and subsequently centrifuged at 21,000× *g* for 30 min at 4 °C. Two hundred and sixty µL of the aqueous phase was transferred to a new 1.5 mL tube, and precipitated with 1/10 (vol/vol) of 3 M sodium acetate (AM-9740, Ambion/Thermo Fisher Scientific) and 2.5 vol of ice-cold 100% ethanol (Commercial Alcohols Inc., Brampton, ON, Canada). The samples were placed at −80 °C for 1 h, and then centrifuged at 21,000× *g* for 30 min at 4 °C. The supernatant was discarded, and the total RNA pellet was washed in 1 mL of 75% ethanol and centrifuged again at 21,000× *g* for 20 min at 4 °C. Finally, the total RNA pellet was air-dried for 10 min at room temperature, resuspended in 150 µL of nuclease-free water (Invitrogen/Thermo Fisher Scientific) by pipette and heated at 55 °C for 10 min.

A subsample of the obtained total RNA (40 µg) was treated with 6.8 Kunitz units of DNaseI (RNase-free DNase kit, QIAGEN), with manufacturer’s buffer at a final concentration of 1×, at room temperature for 10 min to eliminate any residual genomic DNA. Afterwards, the samples were purified using the RNeasy Mini kit (QIAGEN) following the manufacturer’s instructions.

Total RNA purity (i.e., A260/280 and A260/230 ratios) was checked by spectrophotometry using a NanoDrop™ One UV–Vis spectrophotometer (Thermo Scientific) after each step of the extraction/clean-up. The column-purified total RNA samples had A260/280 ratios between 2.10 and 2.15, and A260/230 ratios between 1.84 and 2.40. Total RNA integrity was assessed by 1% agarose gel electrophoresis. The column-purified total RNA samples were of high integrity (tight 28S and 18S ribosomal RNA bands, with 28S being approximately twice as intense as 18S).

#### 2.4.2. cDNA Synthesis

Individual first-strand cDNA templates for qPCR were synthesized in 20 µL reactions from 1 µg of DNaseI-treated, column-purified, total RNA using a High-Capacity cDNA Reverse Transcription Kit which includes random primers. All procedures followed the manufacturer’s (Applied Biosystems/Thermo Fisher Scientific, Waltham, MA, USA) instructions.

#### 2.4.3. Quantitative Real-Time Polymerase Chain Reaction (qPCR) Conditions

All PCR amplifications [endogenous control (normalizer) selection, primer quality control (QC) testing, and experimental qPCR analyses] were performed in 13 µL reactions containing 1× Power SYBR™ Green PCR Master Mix (Applied Biosystems/Thermo Fisher Scientific), 50 nM of both the forward and reverse primers, and the indicated cDNA quantity (see below). Amplifications were performed using the QuantStudio™ 6 Flex Real-Time PCR system (384-well format) (Applied Biosystems/Thermo Fisher Scientific). The real-time PCR program consisted of 1 cycle of 50 °C for 2 min, 1 cycle of 95 °C for 10 min, and 40 cycles of 95 °C for 15 sec and 60 °C for 1 min, with fluorescence detection at the end of each 60 °C step, and was followed by dissociation curve analysis.

#### 2.4.4. Primer Design and Quality Control (QC) Testing

Details on GenBank accession numbers, qPCR primer sequences, amplicon sizes, and amplification efficiencies are presented in Appendix A. Many of these primers were designed previously (see references in Appendix A [8,9,31,32,33,34,35,36,37,38,39,40,41,42]); primers new to this study were designed following the same parameters. Briefly, Atlantic salmon cDNA sequences for a given gene were obtained from GenBank. These sequences were then used to perform BLASTn searches of the non-redundant nucleotide (nr/nt) database of NCBI to obtain all of the available cDNA sequences for this gene and closely related sequences (i.e., paralogues/isoforms); a database was created using Vector NTI (Vector NTI Advance 11, Life Technologies). Next, for each gene, multiple sequence alignments were performed for its corresponding cDNA sequences using AlignX (Vector NTI Advance 11.5.4). If single nucleotide polymorphisms (SNPs), sequencing errors or transcript variants were present, these areas were avoided when designing primers. In the case of gene paralogues/isoforms, these alignments identified regions where paralogue/isoform-specific qPCR primers could be designed (i.e., in an area with ≥3 bp difference between them). Primers were designed with a melting temperature (Tm) of 60 °C mostly using Primer3 [43,44,45]; however, for some gene paralogues/isoforms, they were hand-designed to ensure specificity. Most primers are located in the coding sequence (CDS); the amplicon size range was 87–150 bp.

In the current study, QC testing for each primer pair was performed using a cDNA pool generated from the liver of the same 8 asymptomatic fish [week 5 (3 °C)] assessed in Vadboncoeur et al. [22]. Briefly, standard curves were generated (in duplicate) using a 5-point 1:3 dilution series starting with cDNA representing 10 ng of input total RNA. The amplification efficiency [46] for each primer pair was then calculated using the QuantStudio Real-Time PCR Software (version 1.7.2) (Applied Biosystems/Thermo Fisher Scientific). All showed single-product amplification and absence of primer dimer in the NTC, using dissociation curve analysis. Amplicons were electrophoretically separated on 2% agarose gels, and compared with a 1 kb plus ladder (Invitrogen/Thermo Fisher Scientific) to verify that the correct size fragment was being amplified.

#### 2.4.5. Normalizer Gene Selection

Six candidate normalizer genes [60S ribosomal protein L32 (*rpl32*), eukaryotic translation initiation factor 3 subunit D (*eif3d*), elongation factor 1-alpha 1 (*ef1a1*), elongation factor 1-alpha 2 (*ef1a2*), polyadenylate-binding protein 1 (*pabpc1*), and β-actin (*actb*)] were tested. A qPCR analysis using cDNA representing 5 ng of input total RNA from each individual sample (n = 9) per fish condition (n = 27 in total) was performed (in triplicate) for these six normalizer genes. To identify the most stably expressed normalizers, we conducted a geNorm analysis [47] within the qBase+ software package [48] (https://cellcarta.com/genomic-data-analysis) (Biogazelle, Zwijnaarde, East Flanders, Belgium) using the threshold cycle (C_T_) values obtained for each gene from the 27 samples. Based on this analysis, *eif3d* (geNorm M = 0.255) and *ef1a1* (geNorm M = 0.261) were selected as the two normalizer genes for the experimental (i.e., pools and individual sample) qPCR analyses. For each gene tested, mean C_T_ values per fish condition, and geNorm M values are provided in Appendix A.

#### 2.4.6. Experimental qPCR Analyses

To help select the genes of interest (GOIs) to be assessed in the 27 individual liver samples, a preliminary study was first conducted in which transcript expression levels of 76 GOIs (Appendix A) were tested on cDNA pools representing the three fish conditions (i.e., asymptomatic, ‘early’ symptom and ‘advanced’ symptom). To generate the pools, a subsample of each individual cDNA (n = 9) from a given group was included, with each individual contributing equally. These GOIs were selected based on their functional annotations related to lipid synthesis, transport, and degradation (e.g., *acc1b*, *fasb*, *elovl2*, *fabp3a*, *fabp10b*, *cyp7a1b*, *tnxb*), antioxidant capacity (*catc*), apoptosis (*casp3a*, *casp3b*), immunity (*saa5*, *hampa*), and growth (*igf1*, *ghr1*), and/or that they had been previously shown to be dysregulated in fish and mammalian models of fatty liver disease (FLD and NAFLD, respectively) (e.g., *srebp*, *scd*, *chrebp*, *cd36c*, *pparaa*) [40,49,50,51,52,53,54,55]. Based on this analysis, 34 GOIs with expression levels that differed by ≥1.5-fold amongst at least one of the three fish conditions (the exception being *fasb*) (Appendix A) were selected for measurement in the individual samples.

In both the pooled sample screen and the individual sample experimental qPCR analyses, cDNA representing 5 ng of input total RNA was used as template in the PCR reactions. On each plate, for every sample, the GOIs and normalizer genes were tested in triplicate and a no-template control (also tested in triplicate) was included. In the individual sample qPCR study, since expression levels of a given GOI were assessed across two plates, a plate linker-sample (i.e., a sample that was run on both plates) was also included to ensure there was no plate-to-plate variability. The relative quantity (RQ) of each transcript was determined using the QuantStudio Real-Time PCR Software (version 1.7.2) (Applied Biosystems/Thermo Fisher Scientific) relative quantification study application, with normalization to both *eif3d* and *ef1a1* transcript levels, and with amplification efficiencies incorporated. For each GOI, the sample with the lowest normalized expression (mRNA) level was set as the calibrator sample (i.e., assigned an RQ value = 1.0).

### 2.5. Statistical Analyses

All data are reported as means ± standard error of the mean (SEM). The data were analyzed using Rstudio Version 2023.06.2 + 561 R 4.3.1 (R Core Team, 2023, https://www.r-project.org/). All gene expression data were log_10_-transformed before analysis. A threshold *p*-value < 0.05 was set for all statistical analyses. For univariate analysis (i.e., analysis of variance (ANOVA), Kruskal–Wallis and Welch test), a Dixon outlier test [56] was used to identify outliers before statistical analysis was performed using the function Dixon.test in the package ‘*outlier*’ (Version 0.15) in R. Due to significant differences within the group, one sample in the early symptoms group was removed from the analyses. After outlier removal, normality and homoscedasticity were tested using Shapiro–Wilk and Fligner tests, respectively. One-way ANOVA was used to compare the expression of each gene between the groups, as well as to compare lipid class and FA content. When data were non-normally distributed and/or were not homoscedastic, a Kruskal–Wallis test was used. Post hoc multiple comparisons were performed using Tukey’s HSD tests adjusted with the Benjamin–Hochberg method [57] when the data were normally distributed (using the function Tukey.HSD in the package ‘*agricolae*’ (Version 1.3)); otherwise, the non-parametric Gao test [58] was applied using the function gao_cs contained in the package *‘nparcomp’* (Version 3.0).

The software PRIMER v.7 (PRIMER-E Ltd., Auckland, New Zealand) was used for multivariate analysis. Principal Coordinates Analysis (PCoA), permutational multivariate ANOVA (PERMANOVA), and SIMPER (similarity percentage) analyses were conducted on the expression of the 34 transcripts, the lipid class and fatty acid data, and the phenotypic data from Vadboncoeur et al. [22]. All the data were previously standardized, and the homogeneity of multivariate dispersions was examined before and after transformation. These analyses were based on Bray–Curtis similarities of all pairwise comparisons among individuals. The PERMANOVA analysis was performed with 9999 random permutations and fish condition (symptomatic, ‘early’ and ‘advanced’ symptoms) as a fixed factor. The results of the SIMPER analyses were plotted with the parameters that contributed the most to the differences between groups until ~50% of the cumulative contribution. For the PCoA, the first two dimensions were plotted to obtain a projection of the dataset in a biplot accounting for the most relevant variance [59]. Then, a Pearson correlation analysis, matching each factor with the first two dimensions of the PCoA, was constructed to define the most relevant factors in each dimension.

Finally, to identify correlations between the expression of the 34 analyzed genes and the analyzed lipids (Appendix A), a Pearson correlation matrix was constructed using the package ‘*corrplot*’ (Version 0.92) [60] in R. The correlation coefficients between gene expression and phenotypic characteristics, and between lipids and phenotypic characteristics, were also calculated (Appendix A, respectively). All the data were standardized before the analyses.

## 3. Results

### 3.1. Lipid Analyses

There were no significant (*p* < 0.05) differences between fish with ‘early’ and ‘advanced’ symptoms for any of the measured parameters (Table 1). However, when we compared asymptomatic fish vs. fish with symptoms (i.e., the ‘early’ and ‘advanced’ symptoms groups combined), total lipids and triacylglycerols (TAG) were ~50% higher in symptomatic fish, and sterols and acetone mobile polar lipids (AMPLs) were ~30% lower (Appendix A).

Significant differences were found with regard to the levels of myristic acid (14:0), palmitic acid (16:0), and γ-linolenic acid (18:3ꞷ6). The first two FAs were higher (by 24 and 19%, respectively) in the two groups of symptomatic fish, whereas γ-linolenic acid was ~50% lower (Table 1). In addition: (i) the sum of saturated FAs (ΣSAT) was 17% lower in the two symptomatic groups; and (ii) the DHA:EPA ratio was approximately 12% higher in fish with ‘early’ symptoms as compared to asymptomatic fish. Even when significant differences were not found between the three groups, linoleic acid (18:2ω6), tended (0.05 ≤ *p* ≤ 0.10) to decrease in both fish with early and advanced symptoms (by 12%), while oleic acid tended to increase (by ~19%). The unsaturation index, (ΣMUFA + ΣPUFA):ΣSAT, increased by approximately 30%, and PUFA:SAT and phospholipid:sterol (PL:ST) ratios (markers of membrane fluidity) also tended to increase (*p*~0.06; by ~20% and ~45%, respectively) in fish with symptoms (Appendix A).

### 3.2. Gene Expression

For the qPCR analysis, we selected 34 GOIs related to lipid metabolism, inflammatory and the immune response, apoptosis, and growth. The relative expression of 32 of these transcripts was dysregulated by the development of symptoms. Further, the transcript levels of 27 of these GOIs were significantly different in fish with ‘early’ symptoms compared with the asymptomatic fish, and 31 GOIs had significantly different expression levels when asymptomatic and ‘advanced’ symptom groups were compared. Additionally, 11 genes were differentially expressed between fish with ‘early’ vs. ‘advanced’ symptoms (Figure 1, Figure 2 and Figure 3, Appendix A).

With regard to the mRNA levels of FA synthesis-related genes, *acc1b* (Figure 1A) was significantly lower in fish with symptoms (i.e., with ‘early’ and ‘advanced’ symptoms) vs. asymptomatic fish, while *fasb* showed a trend (*p* = 0.07, Figure 1B) towards downregulation in fish with ‘advanced’ symptoms compared with asymptomatic fish. The expression levels of *scdb* (Figure 1C), a marker of FA desaturation, increased significantly in the fish with symptoms. However, the FA elongation biomarkers *elovl2* (Figure 1D) and *elovl5b* (Figure 1E) were downregulated in the symptomatic fish. A significant downregulation of *elovl5b* was also observed between the ‘early’ and the ‘advanced’ symptom groups. The relative expression of *srebp2* (Figure 1F), a gene involved in sterol synthesis, was significantly upregulated in fish with symptoms, but no differences were detected between the two symptomatic groups. The GOIs related to lipid transport (*fabp3a*, *fabp10b*, *cd36c*, Figure 1G–I) and FA oxidation (*acox3*, *cpt1.2 cyp7a1b*, Figure 1J–L) were all downregulated in fish with ‘advanced’ symptoms compared with the asymptomatic fish. The relative expression of genes related to lipid transport was also lower in fish with ‘early’ symptoms than in the asymptomatic fish, and this was also the case for the lipid oxidation biomarker *acox3*. However, no differences were found for *cpt1.2* and *cyp7a1b* expression between fish with ‘early’ symptoms and those that were asymptomatic.

The mRNA levels of the lipid metabolism-relevant transcription factor *pparaa* (Figure 2A) increased significantly in fish with symptoms, whereas those of *pparab* and *pparga* (Figure 2B,C, respectively) decreased. A significant downregulation was also found in the expression levels of *pparga* in the ‘advanced’ symptom group compared with the ‘early’ symptom group. The transcript levels of the carbohydrate metabolism-relevant transcription factor *chrebpa* did not vary significantly amongst the three groups (*p* = 0.15, Figure 2D). In contrast, the relative expression of the transcription factor *cebpa* (Figure 2E) increased significantly with the progression of symptoms (i.e., ‘advanced’ symptoms > ‘early’ symptoms > asymptomatic). A similar trend was seen for *tnxb* (Figure 2F), a marker of adipocyte homeostasis; however, in this case, the difference in expression levels between fish with ‘early’ and ‘advanced’ symptoms was not significant.

In this study, we detected significant differences in the transcript expression levels of five markers of inflammation when comparing fish with symptoms with those that were asymptomatic. The expression of three of these genes (*5loxa*, *atf6* and *mtor*; Figure 2G,I,K, respectively) was higher in fish with ‘advanced’ symptoms than in the asymptomatic fish. However, there were no significant differences in *5loxa* mRNA levels between the ‘early’ symptom group and the asymptomatic group. In contrast, *atf6* and *mtor* showed significantly higher expression levels in fish with ‘early’ symptoms than in the asymptomatic fish. The transcript levels of *sirt1b* and *pgds* (Figure 2H,J, respectively) were downregulated in fish with ‘advanced’ symptoms compared with asymptomatic fish, and the expression of *pgds* was also significantly lower in fish with ‘early’ symptoms than in the asymptomatic fish.

All five immune response-related GOIs were significantly dysregulated in the fish with symptoms (Figure 3A–G). Two of the genes (*sacs* and *igma*; Figure 3 A,B, respectively) were downregulated in both groups of fish with symptoms as compared with the asymptomatic fish, with no significant differences in expression level between the fish with ‘early’ and ‘advanced’ symptoms. Conversely, the relative expression of *hampa*, *saa5*, and *lect2* (Figure 3C–E, respectively) was upregulated in the fish with symptoms. The transcript levels of *saa5* were strongly upregulated in the fish with symptoms compared with the asymptomatic fish, and this induction was proportional to the symptom’s progression. The mRNA levels of *hampa* and *lect2* were also upregulated, but there was no significant difference between the ‘early’ and ‘advanced’ symptom groups.

The expression of both markers related to oxidative stress response (*prx6* and *catc;*
Figure 3F,G, respectively) was significantly decreased in fish with symptoms. While no difference in *catc* expression was found between fish with ‘early’ and ‘advanced’ symptoms, the expression of *prx6* was significantly lower in fish with ‘advanced’ symptoms than in fish with ‘early’ symptoms. The relative expression of the two selected apoptosis markers was significantly affected by the development of symptoms. For *casp3a* (Figure 3H), an intense upregulation was observed in the fish with symptoms compared with the asymptomatic fish, with the ‘advanced’ symptom group having significantly higher transcript levels than the ‘early’ symptom group. On the other hand, *casp3b* (Figure 3I) showed the completely opposite expression pattern (i.e., downregulation in the fish with symptoms that was proportional to the level of symptom progression).

Finally, the transcript levels of the growth marker *ghr1* (Figure 3J) only increased in the ‘early’ symptom group compared with the asymptomatic group, with no significant difference observed between the asymptomatic fish and the fish with ‘advanced’ symptoms. In contrast, the expression of *igf1* (Figure 3K) decreased progressively as the symptoms worsened.

### 3.3. Multivariate Approach

The PCoA and PERMANOVA based on the relative expression of the 34 genes, the lipid class and the FA profile data, and several phenotypic characteristics (including morphometric and plasma parameters) were able to clearly segregate the three groups of fish (Figure 4A,B, *p*-perm = 0.0001). The top ten factors that contributed the most to dimension 1 (which explained 66% of the total variation) were the genes *igf1*, *acox3*, *cebpa*, *cd36c*, *casp3b*, *lect2*, *catc*, *pparab*, the fish’s score, and HSI (Appendix A). Dimension 2 (which explained 9.2% of the total variation) was driven by the genes *ghr1*, *pparga*, *atf6*, and *fasb*; the plasma concentrations of AST, CK, LDH, Cl^−^, and DHA content; and the DHA:EPA ratio (Appendix A). When comparing asymptomatic fish with those with ‘early’ symptoms using SIMPER analysis, the major contributors to dissimilarity between the groups (average dissimilarity 15.02%) were *saa5*, *casp3a*, plasma [cortisol], *lect2*, *pparga*, fish score, *tnxb*, *igf1*, *catc*, *igma*, *fabp10b*, plasma AST activity, *sacs*, and *cyp7a1b*; with the first four, fish score and *tnxb* being higher in fish with early symptoms, and *pparga* and the last seven being lower in this group of fish (Figure 4C, *p*-perm =0.00004). When comparing asymptomatic fish vs. fish with ‘advanced’ symptoms, the major contributors to the dissimilarity between the groups (dissimilarity 18.69%) that were higher in fish with ‘advanced’ symptoms were *saa5*, *casp3a*, fish score, *lect2*, plasma [cortisol], *tnxb*, and plasma LDH activity, whereas *cyp7a1b*, *pparab*, *igf1*, *catc*, *sacs*, and *elovl5b* were higher in asymptomatic fish (Figure 4D, *p*-perm =0.00002). Finally, when comparing fish with ‘early’ and ‘advanced’ symptoms of liver disease, the major contributors to the dissimilarity between these groups (average dissimilarity 10.29%) were *cyp7a1b*, *pparga*, *igf1*, *elovl5b*, *saa5*, plasma AST activity, *casp3a*, *tnxb*, plasma LDH and CK activity, *5loxa*, fish score, plasma [cortisol], and liver triacylglycerol levels, with the first four factors being significantly higher in fish with ‘early’ symptoms (Figure 4E, *p*-perm =0.0006). Interestingly, *saa5* was the highest contributor to the dissimilarity between the asymptomatic vs. ‘early’ symptomatic fish, the asymptomatic vs. ‘advanced’ symptom comparison, and the second highest contributor to the ‘early’ vs. ‘advanced’ symptom comparison. Further, *casp3a* was the second most important factor contributing to the dissimilarity between the asymptomatic vs. ‘early’ symptomatic fish, and the asymptomatic vs. ‘advanced’ symptom groups.

### 3.4. Correlation Analyses

The Pearson correlation matrix, based on the RQ values of the analyzed genes and the lipid class and FA levels (Figure 5, Appendix A), showed that 820 of the 1054 analyzed correlations were statistically significant. Liver total lipids (TL) and triacylglycerol (TAG) had a similar pattern of correlation with the analyzed GOIs, being positively correlated with ten GOIs (i.e., *scdb*, *srebp2*, *pparaa*, *cebpa*, *5loxa*, *mtor*, *hampa*, *saa5*, *lect2*, and *casp3a*), and negatively correlated with 12 GOI (i.e., *acc1b*, *fabp10b*, *cd36c*, *acox3*, *cyp7a1b*, *pparab*, *pgds*, *chrebpa*, *prx6*, *catc*, *casp3b*, and *igf1*). Conversely, sterols and AMPL showed the opposite correlation pattern, being positively correlated with *acc1b*, *elovl2*, *elovl5b*, *fabp3a*, *fabp10b*, *cd36c*, *acox3*, *cyp7a1b*, *pparab*, *pparga*, *chrebpa*, *sirt1b*, *sacs*, *igma*, *prx6*, *catc*, *casp3b*, and *igf1*, and negatively correlated with *scdb*, *srebp2*, *pparaa*, *cebpa*, *tnxb*, *5loxa*, *atf6*, *mtor*, *hampa*, *saa5*, *lect2*, *casp3b*, and *ghr1*. Free FAs (FFA) and phospholipids (PL) showed a similar pattern to sterols, but with much lower correlation coefficients. Lastly, the PL:ST ratio showed a positive correlation with five GOI (*srebp2*, *tnxb*, *atf6*, *casp3a*, and *ghr1*) and a negative correlation with 15 GOI, including *elovl2*, *srebp2*, *atf6*, *prx6*, and *igf1*.

The sum of SAT (ƩSAT), and myristic (14:0) and palmitic (16:0) acid, were positively correlated with *acc1b*, *cd36c*, *acox3*, *sirt1b*, *prx6*, and *casp3b*, and negatively correlated with *scdb*, *pparaa*, *5loxa*, *mtor*, and *lect2*. With regard to the sum of MUFAs, and vaccenic (18:1ꞷ7), and oleic acids (18:1ꞷ9), they were significantly correlated with 29, 31, and 28 of the evaluated genes, respectively, including positive correlations with *scdb* and *pparaa*, *cebpa*, *saa5*, and *lect2*, and negative correlations with *elovl5b*, *fabp10b*, *chrebpa*, *sacs*, and *igf1*. α-linolenic acid (18:3ꞷ3) content was positively correlated with 12 GOI (e.g., *scdb*, *pparaa*, *chrebpa*, *pgds*, and *lect2*) and negatively correlated with the expression of 21 of the evaluated genes (e.g., *acc1b*, *elovl5b*, *fabp3a*, *acox3*, and *igf1*). In contrast, γ-linolenic acid was less strongly correlated with 24 GOI, including *elov5b*, *mtor*, and *sacs*. Total PUFAs were positively correlated with only three GOI (*cebpa*, *mtor*, and *ghr1*), and negatively correlated with nine GOI, including *elovl5b*, *cd36c*, *pparab*, *5loxa*, and *prx6*. The sum of ω3 FAs showed a similar correlation pattern to PUFAs, but ω6:ω3 showed the opposite pattern. EPA was positively correlated with only seven GOI (e.g., *fabp10b*, *cherbpa*, *sacs*, and *igma*), and negatively correlated with nine GOI (e.g., *pparaa*, *cyp7a1b*, *5loxa*, *mtor*, and *lect2*), whereas ARA had a positive correlation with 7 GOI including *srebp2*, *tnxb*, *atf6*, *igma*, and *ghr1*, and a negative correlation with 17 GOI including *elovl2*, *fabp3a*, *cd36c*, *prx6*, and *catc*. The PUFA:SAT and DHA:EPA ratios showed similar correlation patterns, being positively correlated with 11 GOI, including *srebp2*, *cyp7a1b*, and *ghr1*, and negatively correlated with 17 GOI, including *acc1b*, *cd36c*, *pparga*, *prx6*, and *catc*. Lastly, the unsaturation index [(ƩMUFA + ƩPUFA): ƩSAT] was positively correlated with 18 GOIs, including *acc1b*, *fabp3a*, and *pparga*, and negatively correlated with only five GOIs, including *pparaa*, *hampa*, and *ghr1*.

The Pearson correlation matrix for the relative expression of the GOIs vs. the phenotypic characteristics (Figure 6, Appendix A) showed that there were 495 significant correlations of the 544 tested. Fish score, liver weight, HSI, plasma [lactate], [cortisol], [K^+^], [Na^+^] and [Cl^−^], osmolality, and the plasma markers of tissue damage (AST, LDH and CK activity) showed a similar correlation pattern with the GOIs, but opposite to that observed for weight, length, and plasma [glucose]. The highest correlation values were observed for fish score and HSI, which were negatively correlated with 20 of the evaluated GOIs (in particular *acc1b*, *elovl5b*, *fabp10b*, *cd36c*, *acox3*, *pparab*, *prx6*, *catc*, *casp3a*, and *igf1*), and had positive correlations with 12 of the evaluated genes including *scdb*, *pparaa*, *cebpa*, *5loxa*, *mtor*, and *lect2*. Highly negative correlations were also observed between plasma [cortisol] and *acc1b*, *pparab*, *pgds*, *prx6*, *casp3b*, and *igf1*, whereas positive correlations were found with *cebpa*, *tnxb*, *mtor*, *hampa*, *lect2*, and *casp3a.* Interestingly, the condition factor (CF) was only positively correlated with *cyp7a1b* and *pparga*, and negatively correlated with *ghr1*, and *ghr1* expression was only correlated with liver weight, HSI, CF, plasma [Cl^−^], and osmolality.

Finally, the Pearson correlation matrix for liver lipid composition and phenotypic characteristics (Figure 7, Appendix A) revealed that 287 of the 496 correlations tested were significant. Liver TL and TAG showed a similar correlation pattern; being positively correlated with 11 and 10 phenotypic characteristics, respectively (including fish score, HSI, plasma [lactate], and [cortisol], and plasma AST, LDH, and CK activity), and negatively correlated with plasma [glucose]. Liver TL also had a negative correlation with fish weight and length. In contrast, FFA, sterols, and AMPL showed the opposite pattern, with a positive correlation with fish weight and length, and a negative correlation with fish score, HSI, and plasma [K^+^] and [Na^+^]. No positive correlations between PL and the evaluated phenotypic characteristics were observed, but this lipid class was negatively correlated with 12 phenotypic characteristics including fish score, HSI, plasma [Cl^−^], osmolality AST, LDH, and CK activity.

Myristic and palmitic acids (14:0 and 16:0, respectively) were positively correlated with 4 and 2 of the evaluated phenotypic characteristics, respectively, including plasma [glucose], and were negatively correlated with 9 and 11 phenotypic characteristics, including fish score, HSI, and plasma [cortisol]. A significant correlation was also found between plasma AST activity and palmitic acid, but not with myristic acid. The ƩSAT had a negative correlation with 11 of the evaluated parameters (e.g., fish score, HSI, and plasma [cortisol] and AST activity) but was only positively correlated with plasma [glucose], and this pattern was opposite to that found for the ƩMUFA and oleic and vaccenic acids. In contrast, the ƩPUFAs were only positively (and weakly) correlated with plasma [cortisol], and negatively correlated with CF and plasma [K^+^]. Linoleic acid (18:2ꞷ6) was only significantly correlated with three of the evaluated characteristics, being positively correlated with plasma [K^+^], and negatively correlated with plasma [cortisol] and osmolality. Further, no correlations were observed for γ-linolenic acid. However, α-linolenic acid had strong positive correlations with 11 characteristics, including fish score, HSI, plasma [cortisol], [K^+^], and AST, LDH, and CK activity, and a negative correlation with fish weight, length, and plasma [lactate]. Total liver ω3 FAs, EPA, DPA, and DHA were positively correlated with plasma [cortisol], and negatively correlated with HSI, plasma [K^+^], and AST, LDH, and CK activities; however, none of them were particularly strong. In contrast, total ω6 FAs showed a weak positive correlation with plasma AST, LDH, and CK activity, but also with plasma [cortisol], [lactate], and [K^+^], and fish score. With regard to lipid ratios, DHA:EPA and PUFA:SAT showed a similar correlation pattern, being positively (but weakly) correlated with six and eight characteristics, respectively (e.g., fish score, HSI, and plasma [cortisol]), and negatively correlated with three characteristics, including plasma [glucose]. The unsaturation index [(ƩMUFA + ƩPUFA): ƩSAT] was negatively (and weakly) correlated with only three characteristics (e.g., liver weight, plasma [Cl+], and osmolality), and no positive correlations were found.

## 4. Discussion

Prolonged cold exposure represents a threat to the Atlantic salmon aquaculture industry in eastern Canada, Iceland, and northern Norway based on reported mortalities [11,12,13,14,15,16] and the physiological [61,62] and immunological disturbances [28,63] that are known to be associated with cold temperatures in fishes. To improve the sustainability of this industry, it is essential to confirm the susceptibility of Atlantic salmon to WS/WD and FLD at these temperatures, as suggested by Vadboncoeur et al. [21,22], and improve our understanding of the underlying mechanisms that trigger the development of this disease. In this study, we collected hepatic lipid class, FA, and transcript expression data on healthy (asymptomatic) fish vs. those identified as having symptoms of WS/WD [22], and correlated these values with phenotypic data for these same fish. We report that: (i) fish with symptoms of WS/WD had livers with ~50% higher levels of total lipid and TAG; (ii) 32 of 34 genes previously identified as related to FLD in fishes and/or NAFLD in mammalian models were dysregulated in these fish; (iii) multivariate statistical models were able to clearly separate asymptomatic (healthy) fish from those with ‘early’ vs. ‘advanced’ symptoms; and (iv) the majority of variation between these groups could be explained by a few biochemical and molecular markers. Thus, this study clearly establishes that FLD can occur in Atlantic salmon at cold temperatures and provides key information that may assist the industry in addressing the loss of fish in the winter months.

The fish used in these experiments had been held at ~3 *°*C for more than 5 weeks, and while some salmon had already been culled/euthanized or developed symptoms of WS/WD (~30%), the majority of fish appeared to be healthy. However, once the initial (‘early’) symptoms of WS/WD appeared in a particular fish (i.e., lethargy, erratic swimming), the progression of this syndrome/disease was extremely rapid, and fish became moribund in ≤1 week [22]. This rapid progression of WS/WD is consistent with that observed for other fish species [61], and has been associated with the direct and indirect (i.e., starvation/dramatically reduced feed intake) effects of prolonged cold exposure. However, the healthy fish in this study continued to eat at 3 *°*C, and thus, we believe that the symptoms exhibited by the salmon in this study were solely related to long-term exposure to this temperature. That low temperature alone can result in WS/WD is consistent with other studies on fish [28,64].

FLD in fishes (NAFLD in mammals) is thought to be a manifestation of a hepatic metabolic syndrome, the initial step of which is excessive accumulation of TAG in the liver. This hepatic steatosis is then followed by a pathogenic state called steatohepatitis (FLD) that is the result of a sequence of events (the release of inflammatory cytokines, oxidative stress, mitochondrial dysfunction, endoplasmic reticulum (ER) stress, and finally, apoptosis) [55,65,66]. In the following discussion, we compare our lipid and gene expression data with those reported in studies that have used various approaches to induce hepatic steatosis (fatty liver) and FLD in fishes, and use these data to substantiate that these Atlantic salmon held at 3 *°*C did indeed develop FLD.

### 4.1. Cold Exposure and Hepatic Steatosis (Fatty Liver)

#### 4.1.1. Increases in Liver Lipids and Triacylglycerols

The symptomatic fish in this study had enlarged, pale, and friable livers [21,22], with ~50% higher levels of total lipid and TAG, as compared to asymptomatic fish (Table 1; Appendix A). Prolonged cold exposure has been shown to cause hepatic lipid deposition in many fishes, including gilthead sea bream (*Sparus aurata*), yellowtail kingfish (*Seriola lalandi*), spotted sea bass (*Lateolabrax maculatus*), largemouth bass (*Micropterus salmoides*), sea trout (*Salmo trutta trutta*), and catfish (*Hoplosternum littorale*) [28,50,51,67,68,69]. With regard to the Atlantic salmon, several studies have examined the effects of cold temperatures on liver size and lipid composition. For example, Vadboncoeur et al. [21] showed that salmon given a slow temperature decrease from 8 to 1 °C, and held at 1 °C for an additional week, had higher HSI values (by ~35%) as compared to fish held at 8° C. Dessen et al. [29] reported that cage-reared salmon that became moribund at 5 °C after being switched onto a feed with a higher lipid content had 25% greater hepatic lipid levels. Finally, Sissener et al. [70] reared salmon at 6 and 12 °C and found that salmon reared at the lower temperature had 2–7-fold higher levels of hepatic TAG. However, rearing at low temperatures does not always result in the above changes. In contrast to Sissener et al. [70], Ruyter et al. [71] held salmon at 5 vs. 12 °C, and did not find any difference in liver lipid content when fed a diet which predominantly contained fish oil vs. plant-based oils.

Triacylglycerol accumulation in the liver at cold temperatures could occur due to several factors, including: (i) an increase in the uptake of FAs from the blood, and their deposition in hepatocytes as TG (i.e., de novo lipogenesis [DNL]); (ii) a decrease in the use of FAs in the liver for energy production (i.e., via β-oxidation); and/or (iii) a reduction in the production of very low-density lipoproteins (VLDLs) and their subsequent export from the hepatocytes [55,72]. In fish, the mobilization of FAs from muscle and perivisceral muscle stores due to stress (i.e., high circulating cortisol levels), and their subsequent uptake by hepatocytes, is thought to be a major contributor to FLD [55,67,73,74].

We did not measure plasma-free FAs in this study, but [22] reported that symptomatic fish had circulating cortisol levels indicative of severe stress (i.e., >100 ng·mL^−1^), and it is expected that such levels of this corticosteroid would have mobilized energy stores (incl. FAs) from the muscle and perivisceral tissues [25,64,74]. These FAs would then have to be transported into the hepatocytes before being resynthesized into TG. Thus, it was somewhat surprising that the expression of *cd36c* (a FA translocase) was significantly downregulated in both groups of symptomatic fish (Figure 1I) as it is well known that CD36 increases FFA uptake in the liver [75]. However, upregulation of this gene is generally associated with the onset of hepatic steatosis, and the HSIs of fish with ‘early and ‘advanced’ symptoms were both ~2-fold greater than in asymptomatic fish. Further: (i) no increase in *cd36* expression was seen in Atlantic salmon primary hepatocytes treated with oleic acid, which led to considerable lipid deposition [76]; (ii) the downregulation of hepatic *cd36* diminishes FFA uptake, and increases β-oxidation and autophagy, protecting against steatohepatitis; and (iii) there is some evidence that CD36 deficiency might contribute to steatohepatitis by an FA-independent mechanism [75,77]. That the liver/hepatocytes of the symptomatic fish had transitioned from the stage of hepatosteatosis to steatohepatitis is consistent with the expression profile of a number of other examined genes. For example, the expression of two key genes related to FA synthesis (*acc1b* and *fasb*, Figure 1A,B) was downregulated and unchanged, respectively, in asymptomatic vs. symptomatic fish, and *pparga* and *chrebpa* (which stimulate the uptake of FAs and drive hepatic DNL, Figure 2C,D) were also downregulated or not different in symptomatic fish.

To this point, we have focused on how the changes in liver gene expression related to FA uptake and DNL were impacted by the development of symptoms of WS/WD. However, lipid and TG accumulation with FLD can also be due to a dramatic reduction in lipid catabolism. Our results for asymptomatic vs. symptomatic salmon are consistent with this paradigm. Acetyl-CoA oxidase (ACOX) is the first enzyme in the peroxisomal β-oxidation pathway, carnitine palmitoyltransferase (CPT1) is an essential step in the beta-oxidation of long-chain FAs by the mitochondria [78], and transcript expression of both of these key modulators of β-oxidation was downregulated in symptomatic fish (Figure 1J,K). Further, FABP3 has been shown to have a high binding affinity for 18:1 n-9 and n-6 FAs, and is believed to transport these FAs into the mitochondria for β-oxidation [79], and the transcript expression of this gene was also downregulated greatly in symptomatic fish (Figure 1G). This likely contributed to the development of FLD in this study. In contrast, it is not possible to specifically interpret how changes in PPARα influenced FA β-oxidation (or inflammation through the regulation of NF-κB [80]). Besides its lipid metabolism regulation roles, this transcription factor also induces the expression of the proinflammatory mediators, tumor necrosis factor-a (TNFα) and interleukin 6 (IL-6) [81]. While the existence of two paralogues of PPARα has been confirmed in salmonids [82], to our knowledge, no information on their specific role(s)/function(s) in the fish liver is/are available. Also, the response of these two transcription factors to the development of fatty liver was completely opposite. The expression of *pparaa* increased in fish with ‘early’ and ‘advanced’ symptoms, whereas that of *pparab* decreased by approximately the same amount (Figure 2A,B). These paralogues could play antagonistic roles or have different functions due to the salmonid-specific gene duplication event, as reported for the genes *mmp19* and *ccl19* in Atlantic salmon [83]

#### 4.1.2. Changes in Liver Lipid Classes and Fatty Acids

In this study, we also found changes in the levels of specific lipid classes and FAs between asymptomatic and symptomatic fish (Table 1, Appendix A). Symptomatic fish had lower levels of sterols and acetone mobile polar lipids (AMPLs) (by 34 and 38%, respectively) than the asymptomatic fish. That sterols went down is not consistent with other studies which have shown that liver cholesterol increases greatly when fatty liver is induced in fishes by a number of experimental manipulations (e.g., low temperature [75]; high-fat diet [84]; high starch diet [85]), and was unexpected given the changes in the transcript expression of both *srebp2* and *cyp7a1b* (Figure 1F,L). Sterol regulatory element binding protein 2 (SREBP2) is a transcription factor that controls cholesterol homeostasis by stimulating the expression of sterol-regulatory genes, and transcript expression of this gene was higher in both symptomatic groups (Figure 1F). Further, 7α-hydroxylase (CYP7A1) is the rate-limiting enzyme in the classic bile acid synthesis pathway, and the transcript expression of this gene was substantially lower in fish with ‘advanced’ symptoms (Figure 1L). However, not all models of fatty liver disease in fishes have shown an increase in *srebp2*. Espe et al. [76] reported that srebp2 transcript expression was not different in Atlantic salmon cultured hepatocytes when exposed to elevated levels of oleic acid that induced a 6-fold increase in intracellular lipid droplets. To our knowledge, there are no specific data on the effects of FLD on AMPL, although this class of lipids has been shown to vary between demersal and pelagic fish species [86].

We also found that myristic and palmitic acid proportions and ƩSAT decreased by ~20%, and that the unsaturation index (ΣMUFA+ ΣPUFA:ΣSAT) increased by 30% in fish with symptoms of FLD, but that there was no difference in ƩPUFAs (Table 1; Appendix A). This could be partially due to the fact that the Atlantic salmon tends to protect long-chain FAs from oxidation by consuming saturated ones, especially palmitic acid [87]. However, the transcript expression data for asymptomatic vs. symptomatic fish provide insights into other potential mechanisms mediating this response. Delta 9 acyl-CoA desaturase (*scdb*) transcript levels were higher in both symptomatic groups (Figure 1C), suggesting an increase in the number of FAs converted into monounsaturated FAs (e.g., oleic acid). Further, both *elovl2* and *elolvlb* (which are acyl elongases; Figure 1D,E) were significantly downregulated in symptomatic fish. Specifically, in Atlantic salmon, ELOLV5B elongates C_18_ and C_20_ PUFA, with low activity towards C_22_ PUFA, whereas ELOVL2 elongates C_20_ and C_22_ PUFA, with lower activity towards C18 PUFA [88]. Studies using liver microsomal protein from wild-type and knockout mice have also demonstrated that the elongation of γ-linolenic acid (18:2ꞷ6) to dihomo-γ-linolenic acid (20:3ꞷ6) and stearidonic acid (18:4ꞷ3) to ω3-eicosatetranoic acid (ARA, 20:4ꞷ3) requires ELOVL5 activity [89]. Based on the above changes in gene expression it is not surprising that ƩSAT, myristic, and palmitic acids (saturated FAs) were lower in symptomatic fish. However, we would have also expected that the DHA:EPA would have decreased (not increased) based on the lower transcript expression levels of *elovl2*. Thus, it is possible that either *elovl2* transcript expression did not reflect protein expression or that other factors were at play. For example, the selective retention of DHA could be in an effort to enhance membrane fluidity or to increase the synthesis of eicosanoid compounds (the latter suggested by the increase in *pgds* expression). With regard to changes in FA composition, it is very interesting that γ-linolenic acid increased, and that oleic acid (18:1ꞷ9) increased despite the decrease in *scdb* transcript expression. However, these may be key findings as higher γ-linolenic acid levels in the culture media of human liver cells significantly increase cell death after 12 h of incubation [90], and oleic acid has been shown to induce FLD in primary cultures of Atlantic salmon hepatocytes [76]. Importantly, the changes in saturated FAs, monounsaturated FAs, and oleic acid in this study are consistent with other studies on the effects of cold temperatures. These studies consistently show a decrease in saturated FAs in the liver, and increases in various monounsaturated FAs including oleic acid content [29,71].

### 4.2. Evidence of Steatohepatitis (Fatty Liver Disease) in Symptomatic Fish

In addition to the accumulation of triacylglycerols, and changes in the lipid class/FA composition, FLD in fish and mammals is characterized by mitochondrial dysfunction, oxidative stress, the release of inflammatory cytokines, endoplasmic reticulum (ER) stress, and apoptosis [55,65,66]. The accumulation of hepatic free FAs initially results in a metabolic shift to overcome the hepatic FFA burden, and this involves increased TCA cycle stimulation and mitochondrial FA oxidation (FAO). A major consequence of the increase in FAO is an increase in the supply of reducing equivalents to the electron transport chain (ETC) and in ROS production (oxidative stress) [91]. We did not measure ROS production or markers of oxidative stress in this study (i.e., lipid, protein, and DNA damage); however, we did measure the expression of genes related to antioxidant function (e.g., *prdx6* and *catc*; Figure 3F,G). The diminished expression and activity of ROS detoxification mechanisms (e.g., SOD, catalase or GSH) with steatosis have also been reported in mammalian in vitro and in vivo experiments [92]. It has been suggested that oxidative damage associated with NAFLD may be the result of a decrease in the antioxidant defense system, including CAT activity levels [93]. Further, our findings are consistent with Xue et al. [94], who reported that catalase, but not SOD activity, was reduced in largemouth bass that had developed FLD.

The symptomatic fish had higher expression levels of several pro-inflammatory biomarker genes (i.e., *5loxa*, and *igma*; Figure 2G and Figure 3B), the latter indicative of the role that antibodies secreted by B cells play in the enhancement of NAFLD pathology and in inducing adipose tissue dysfunction [52,95]. Further, the expression of *sirt1b*, a protein shown to protect against NAFLD pathogenesis, was reduced in symptomatic fish [55]. In contrast, the expression of *pgds* was actually lower in symptomatic fish (Figure 2J), and that of *mtor* was higher (Figure 2K). PGDS can be either pro- or anti-inflammatory depending on the fatty acid used as substrate, and its expression appears to be negatively linked to the expression of the *pparg* [96]. mTOR regulates several pathways that appear to lessen/inhibit the development of FLD. For example, mTOR blocks FFA formation and DNL, the transcription of INSIG (the key gene determining SREBP production), and can directly regulate various inflammatory factors of the IL family (e.g., TNF-α, NF-ĸB, and JNK), and limit the progression of the FLD [97]. Thus, it appears that the balance between pro- and anti-inflammatory regulatory factors determines whether FLD occurs, and its extent, and that in the salmon in this experiment those that result in FLD dominated. With regard to transcript levels of the above genes’ in fish models of hepatic steatosis and FLD, there are limited data to which our data can be compared. Hammes et al. [98] showed that taurine supplementation reduced hepatic steatosis induced by thioacetamide in zebrafish, and this effect was associated with an increase in sirt1 mRNA expression, in agreement with our study (Figure 2H). In contrast, Jia et al. [89] reported that high fat-induced steatosis in tilapia was not associated with a change in *mtor* expression.

Fatty liver disease is associated with secretory pathway dysfunction, and this results in induction of the unfolded protein response (UPR). Activating transcription factor 6 (ATF6), is one of three main UPR sensors [99]. However, whether it protects against or augments hepatic steatosis/steatohepatitis depends on whether the condition is acute or chronic. For example, it has been shown that steatosis in zebrafish caused by acute stress is augmented by ATF6 loss, but that steatosis caused by chronic stress is reduced when ATF6 is depleted [99,100]. In this experiment, *aft6* expression was strongly upregulated (Figure 2I), and this could be viewed as protecting against or augmenting FLD, respectively. As mentioned above, WS/WD developed very quickly in these salmon (within a week), and given this timeframe, it is difficult to determine the role of ATF6 in this particular model of hepatic steatosis/steatohepatitis. This is particularly true as *atf6* expression was not strongly correlated with either HSI or liver TAG levels (Figure 5 and Figure 6), and Jia et al. [84] did not report significant differences in *atf6* expression in tilapia that developed steatosis after being fed a high-fat diet for up to 90 days.

The accumulation of excess saturated FAs leads to apoptosis through oxidative and endoplasmic reticulum stress [101]. In this study, we measured the transcript expression of caspase 3a and 3b as markers of apoptosis, and the former was greatly upregulated in symptomatic fish, whereas the latter was downregulated (Figure 3H,I). The upregulation of *casp3a* is consistent with the increase in *casp3* expression in hybrid grouper (*Epinephelus fuscoguttatus × Epinephelus lanceolatus*) that developed hepatic steatosis when fish protein was replaced with rendered animal protein [102], and the established role of caspase 3 in the intrinsic (mitochondrial) pathway of cell death [103]. However, the low homology between the caspase-3a and -3b paralogues in Atlantic salmon, medaka (*Oryzias latipes*), and zebrafish (*Danio rerio*), and that these genes are present on different chromosomes [104], suggests that they radiated after an early genome duplication event [103]. Further, the opposite responses of these two genes to WS/WD and FLD suggest that caspase 3b underwent neofunctionalization. At present, there are insufficient data on what the function(s) of caspase 3b might be.

As has been reported for other fish species that developed WS/WD, the latter parts of this condition were associated with head and dermal ulcers characteristic of a bacterial infection (see [21,22]). Thus, we measured a number of genes involved in the fish’s immune response. The transcript expression of *sacs* and *igma* were downregulated, whereas those of *hampa*, *saa5*, and *lect2* were upregulated (Figure 3A–E). The significance of the downregulation of *igma* is discussed above. However, at present, we do not have an explanation for the downregulation of sacsin (*sacs*) transcription. SACSIN is highly upregulated by stress and viral infections [105,106,107], and there is no information on its role in FLD. So, this is the first time that this gene is has been shown to be associated with liver damage in fish. Hepcidin (*hampa*) is an important antimicrobial peptide in teleost fish, and thus, it is not surprising that *hampa* expression was upregulated in symptomatic fish. However, recent research has highlighted HAMP’s ability to upregulate the mRNA expression of both pro- and anti-inflammatory cytokines [108]. Thus, it is possible that *lect2* expression played a significant role in modulating the severity of inflammation in the liver of symptomatic salmon. Like hepcidin, leukocyte-derived chemotaxin-2 (*lect2*) also has direct antimicrobial activity (in addition to its more well-known chemotactic and phagocytosis-stimulating activities [109]), and its expression has been linked to inflammation in the fish liver [110]. Finally, the expression of serum amyloid 5 (*saa5*) was also significantly upregulated in symptomatic fish, with the increase in fish with ‘advanced’ symptoms greater than in those with ‘early’ symptoms (Figure 3D). SAAs (including *saa5*) are well-known acute phase proteins in salmonids and other teleosts that respond to bacterial infection and pro-inflammatory cytokines [111], and whose function includes the recruitment of immune cells to inflammatory sites in an attempt to restore homeostasis [112].

As part of this study, we also examined how WS/WD and FLD impacted the expression levels of two important modulators of growth (*ghr1* and *igf1*). The transcript expression of *ghr1* was elevated in fish with ‘early’ symptoms compared with the asymptomatic fish and those with ‘advanced’ symptoms, whereas the expression of *igf1* decreased as FLD progressed (Figure 3J,K). In a recent review, GH and IGF-1 were identified as having fundamental (synergetic) roles in the pathogenesis of non-alcoholic steatohepatitis (NASH) [113]. Indeed, it was suggested that GH may mediate its protective effect by inducing adipose tissue lipolysis [114], while IGF-1 influences cholesterol transport. IGF-1 has also been shown to have antifibrotic properties [115], including in NAFLD and NASH. Thus, lower expression of *ghr1* and *igf1* likely promoted the development of FLD in the salmon in this study. This conclusion would also be consistent with research on other fish species. Wu et al. [116] showed that common carp (*Cyprinus carpio*) overexpressing growth hormone had reduced lipid deposition even when fed a high-starch diet, and Zeng et al. [117] reported that IGF1-deficient male zebrafish developed fatty liver.

### 4.3. Proposed Biomarkers of FLD in Atlantic Salmon

New tools to investigate the impacts of long-term low temperature exposure on fishes, specifically, Atlantic salmon, are essential for the sustainability of the industry in cold regions because of the substantial mortalities associated with sea-cage rearing in winter. In this regard, we used multivariate (PCoA, PERMANOVA, and SIMPER) and correlation analyses to identify biomarkers of FLD in Atlantic salmon.

According to our PCoA analysis, we found that cold-exposed Atlantic salmon post-smolts that were apparently healthy and fish with symptoms of FLD were clearly separated by dimension 1, and that the factors that contributed the most (>85%) to this dimension (i.e., that grouped with fish with symptoms, marked as blue in Appendix A) were fish score, HSI, and the genes *cebpa* and *lect2*, which are related to the regulation of lipid metabolism and inflammation, respectively. Not surprisingly, fish with high HSI values and high scores also had the highest positive correlation values with *lect2*, *mtor*, and *cebpa*, and the highest negative correlation values with *cd36c*, *casp3b*, *igf1*, and *acox3* (Figure 6, Appendix A). It is worth noting that the gene *lect2* was also one of the main contributors to the differences between asymptomatic fish and both groups of fish with symptoms; explaining ~4% of the difference between these groups according to SIMPER analysis. However, in this analysis, *saa5* (an acute phase protein) and *casp3a* (a marker of apoptosis) contributed the most (~9 to 10 and 5%, respectively) to the difference between asymptomatic fish and those with both ‘early’ and ‘advanced’ symptoms (Figure 4C,D). On the other hand, the PCoA’s dimension 2 mainly explained the difference between fish with ‘early’ and ‘advanced’ symptoms (i.e., the progression of the disease), and here, the genes *cyp7a1b* and *saa5* explained the most dissimilarity; ~7 and 6%, respectively (Figure 4E).

The differences between the contribution to each dimension in the PCoA and the SIMPER analysis could be due to the fact that SIMPER works on high-dimensional relationships, and we are looking at a biplot that does not explain all the variation. Nonetheless, the axes explained >70% of the variation, and identified common markers, which were highly correlated with the appearance of external symptoms [score and enlarged livers (high HSI values)]. Thus, herein, we propose *saa5*, *casp3a*, *igf1*, *acox3 casp3b*, *cebpa*, *cd36c*, *lect2*, and *mtor* as possible biomarkers of FLD in the Atlantic salmon. All of these have been linked to NAFLD or hepatic steatosis in mammalian models [75,97,115,118,119,120,121,122], but surprisingly only a few of them have been linked to FLD in fishes, including *cebpa* in zebrafish [123] and *cd36c* in cultured Atlantic salmon liver cells [76]. This is despite research that has reported changes in the transcript expression of *cd36c*, *cebpa*, *lect2*, and *saa5* in fish with changes in diet or the presence of a bacterial infection [37,52,110,124]. To our knowledge, this is the first study that proposes the above-mentioned genes as markers of FLD in fish. Moreover, we also linked changes in the expression of those genes with prolonged cold exposure-induced stress.

Importantly, our analyses (Vadboncoeur et al. [22] and present study) do not discriminate between whether (1) salmon develop fatty liver (i.e., a metabolic syndrome) when exposed to cold temperatures for prolonged periods, and that this makes them susceptible to bacterial infections; or (2) the salmon became immune-compromised by cold exposure and became susceptible to opportunistic bacterial infections, and that FLD is one of the symptoms/manifestations concomitant with this infection [22].

## 5. Conclusions

In the present study, we analyzed liver lipid content and FA composition, and the expression of 34 genes related to FLD and NAFLD, in Atlantic salmon held at ~3 °C for >five weeks. We report that fish which developed symptoms of WS/WD had an ~50% higher hepatic lipid content (especially triacylglycerols), and that the expression of 32 of the 34 analyzed genes was dysregulated in fish that developed symptoms of WS/WD. Thus, these results provide very strong evidence that salmon held at very low temperatures can develop FLD [22,125]. Finally, while the use of multivariate and correlation analyses allowed us to identify nine potential biomarkers of this disease in Atlantic salmon, the sequence of events that led to its manifestation is still not well understood. Future research must focus on this question so that the aquaculture industry can determine whether the development of vaccines against bacterial pathogens (e.g., *Tenacibaculum* sp.), or of functional diets that prevent FLD (e.g., see [126]), would be the most effective for improving salmon welfare/health, and reducing mortalities, during the winter months.

## Figures and Tables

**Figure 1 biology-13-00494-f001:**
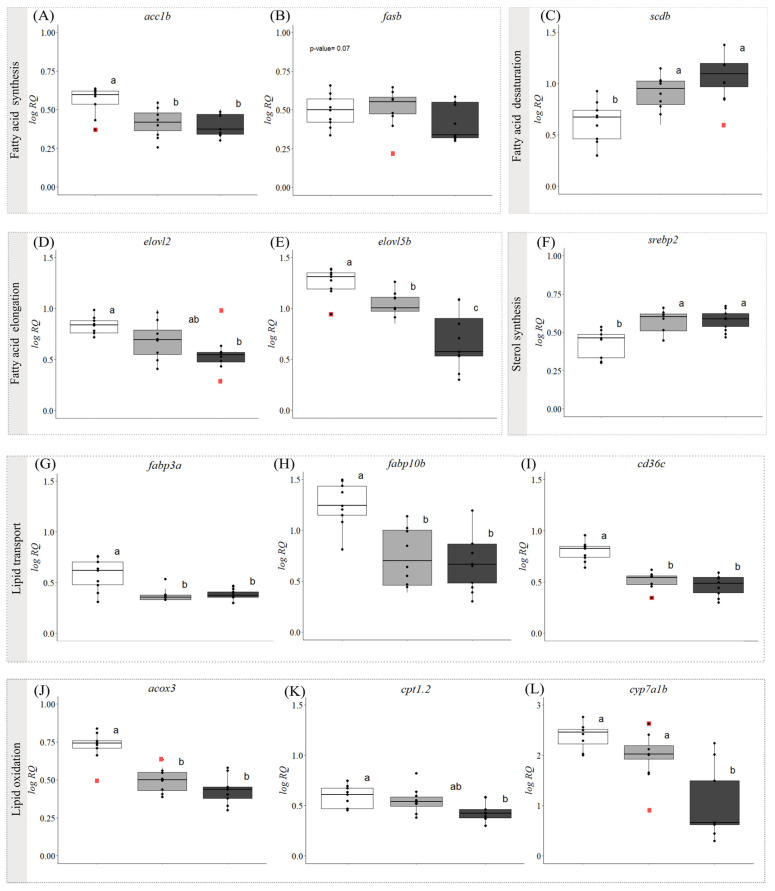
Hepatic transcript expression levels (Log_10_ Relative Quantification) of target genes in asymptomatic fish (white box) and fish with ‘early’ (light-grey box) and ‘advanced’ (dark-grey box) symptoms of fatty liver disease (n = 9 per fish condition). Lower and upper box boundaries indicate the 25th and 75th quartiles, respectively, the line inside the box is the median value, and the vertical lines delimit the 10th and 90th percentiles, respectively. The red dots represent outliers (excluded from statistical analysis). Lowercase letters indicate a significant difference (*p* < 0.05) between groups. The plots are organized according to six different functions. Fatty acid (FA) synthesis: *acc1b* (**A**), *fasb* (**B**). FA desaturation: *scdb* (**C**). FA elongation: *elovl2* (**D**), *elovl5b* (**E**). Sterol synthesis: *srebp2* (**F**). Lipid transport: *fabp3a* (**G**), *fabp10b* (**H**), *cd36c* (**I**). Lipid oxidation: *acox3* (**J**), *cpt1.2* (**K**), *cyp7a1b* (**L**).

**Figure 2 biology-13-00494-f002:**
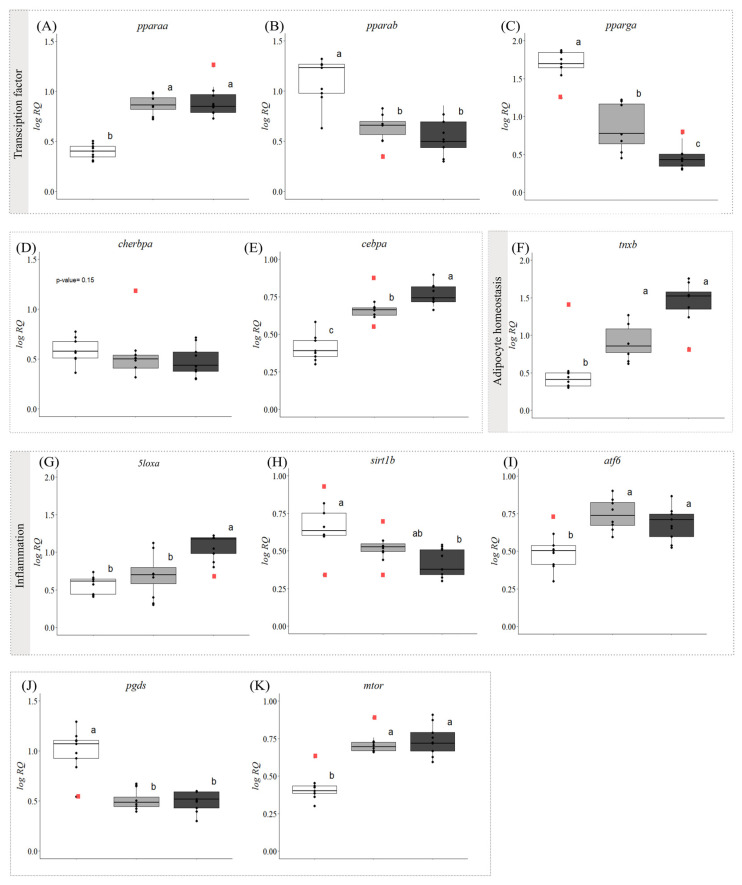
Hepatic transcript expression levels (Log_10_ Relative Quantification) of target genes in asymptomatic fish (white box) and fish with ‘early’ (light-grey box) and ‘advanced’ (dark-grey box) symptoms of fatty liver disease (n = 9 per fish condition). Lower and upper box boundaries indicate the 25th and 75th quartiles, respectively, the line inside the box is the median value, and the vertical lines delimit the 10th and 90th percentiles, respectively. The red dots represent outliers (excluded from statistical analysis). Lowercase letters indicate a significant difference (*p* < 0.05) between groups. The plots are organized according to three different functions. Transcription factors: *pparaa* (**A**), *pparab* (**B**), *pparga* (**C**), *cherbpa* (**D**), *cebpa* (**E**). Adipocyte homeostasis: *tnxb* (**F**). Inflammation: 5*loxa* (**G**), *sirtb* (**H**), *atf6* (**I**), *pgds* (**J**), *mtor* (**K**).

**Figure 3 biology-13-00494-f003:**
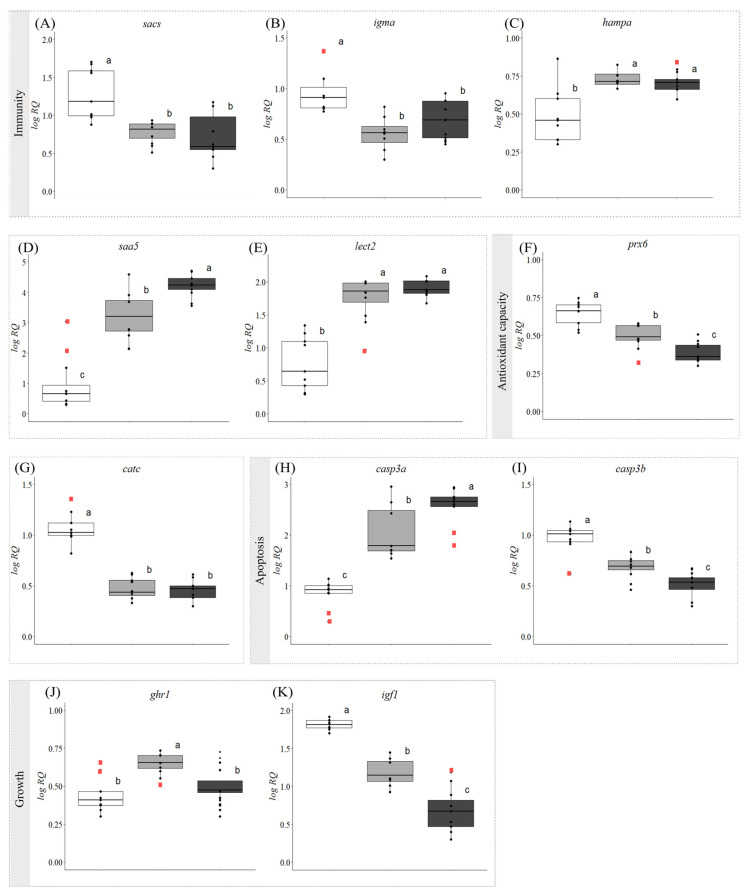
Hepatic transcript expression levels (Log_10_ Relative Quantification) of target genes in asymptomatic fish (white box) and fish with ‘early’ (light-grey box) and ‘advanced’ (dark-grey box) symptoms of fatty liver disease (n = 9 per fish condition). Lower and upper box boundaries indicate the 25th and 75th quartiles, respectively, the line inside the box is the median value, and the vertical lines delimit the 10th and 90th percentiles, respectively. The red dots represent outliers (excluded from statistical analysis). Lowercase letters indicate a significant difference (*p* < 0.05) between groups. The plots are organized according to four different functions. Immunity: *sacs* (**A**), *igma* (**B**), *hampa* (**C**), *saa5* (**D**), *lect2* (**E**). Antioxidant capacity: *prx6* (**F**), *catc* (**G**). Apoptosis: *casp3a* (**H**), *casp3b* (**I**). Growth *ghr1* (**J**), *igf1* (**K**).

**Figure 4 biology-13-00494-f004:**
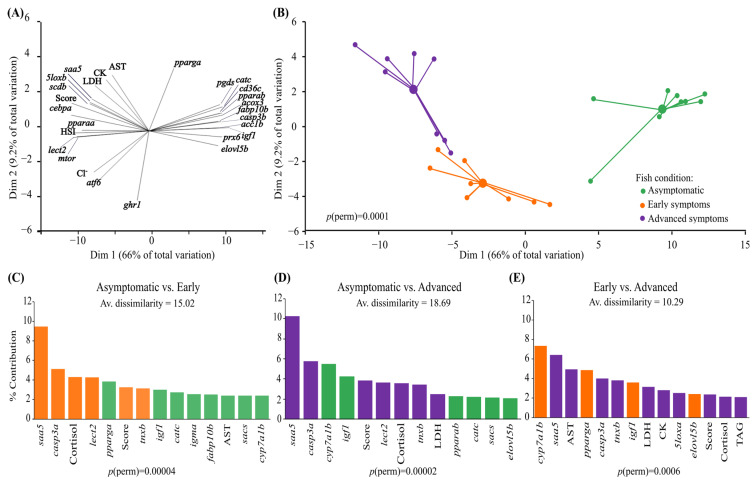
Principal Coordinates Analysis (PCoA), PERMANOVA, and SIMPER analysis between target genes, phenotypic traits, and lipid classes in asymptomatic fish and fish with symptoms of fatty liver disease: (**A**) PCoA plot showing the association (Pearson correlations) of a given biomarker with the coordinates (axes x and y). Only vectors with Pearson correlation coefficients > 0.7 are shown. (**B**) PCoA plot illustrating the differential distribution of the individuals in the two-dimensional space connected to the centroid of each group, with overall PERMANOVA result *p*(perm) = 0.0001. The variance explained by each dimension (PCO1, PCO2) is presented in percentage (%). SIMPER analysis comparing (**C**) asymptomatic vs. fish with ‘early’ symptoms, (**D**) asymptomatic vs. fish with ‘advanced’ symptoms, and (**E**) fish with ‘early’ symptoms vs. ‘advanced’ symptoms shows the contribution of each factor to dissimilarities between groups up to 50% of the total explained dissimilarity. The average dissimilarities between groups are presented in percentage (%) and were validated by a pairwise PERMANOVA. The color code in plots (**C**–**E**) corresponds to the color code used in the PCoA plot (**B**), the bar color represents the group that had the highest contribution in the SIMPER analysis.

**Figure 5 biology-13-00494-f005:**
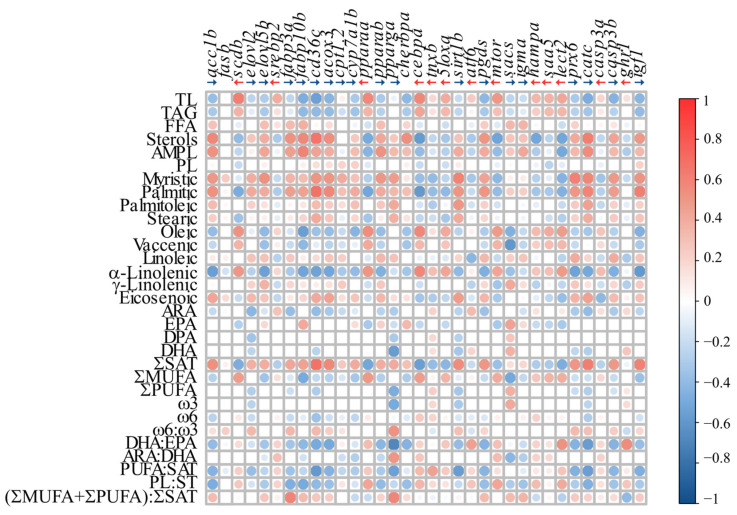
Pearson correlation matrix for asymptomatic fish and fish with symptoms of fatty liver disease (FLD) based on the Log RQ values of the 34 analyzed genes, lipid class composition, and FA content. Positive correlations are in red (+1) and negative correlations are in blue (−1), and the intensity of the color and the size of the dot are in accordance with the correlation coefficient. Non-significant correlations (*p* > 0.05) are not shown. The arrows preceding a gene name indicate whether the gene was upregulated (red) or downregulated (blue) based on the qPCR analysis. When no arrow is present, there was no significant difference in gene expression.

**Figure 6 biology-13-00494-f006:**
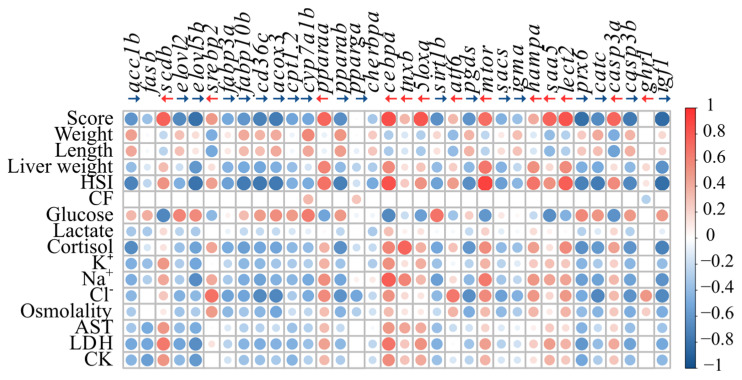
Pearson correlation matrix for asymptomatic fish and fish with symptoms of liver disease based on the Log RQ values of the 34 analyzed genes and the phenotypic traits (data from [22]). Positive correlations are in red (+1) and negative correlations are in blue (−1), the intensity of the color and the size of the dot are in accordance with the correlation coefficient. Non-significant correlations (*p* > 0.05) are not shown. The arrows preceding a gene name indicate whether the gene was upregulated (red) or downregulated (blue) based on the qPCR analysis. When no arrow is present, there was no significant difference in gene expression.

**Figure 7 biology-13-00494-f007:**
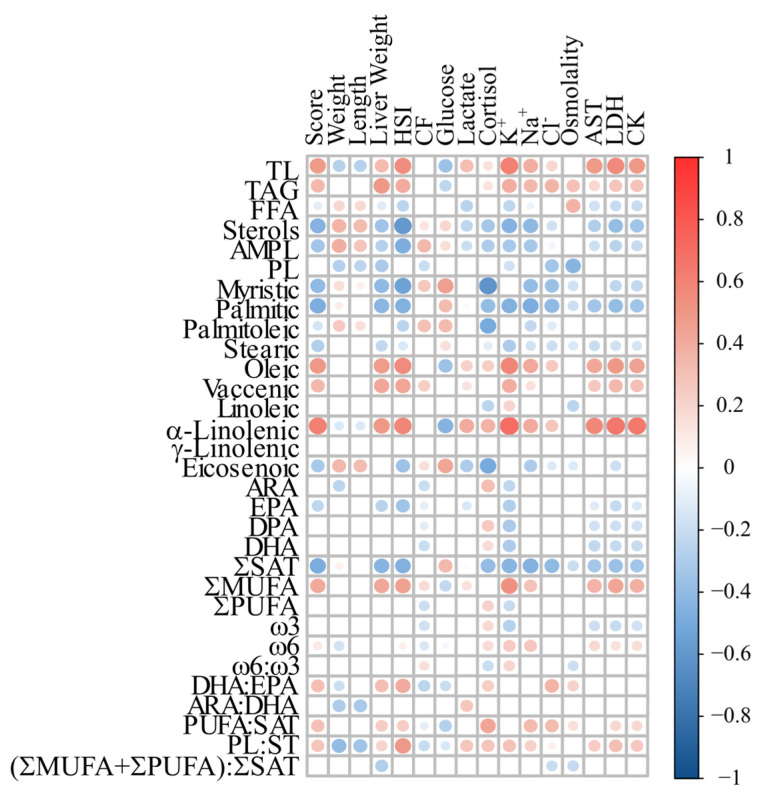
Pearson correlation matrix for asymptomatic fish and fish with symptoms of liver disease based on the lipid classes, FAs, and the phenotypic traits (data from [22]). Positive correlations are in red (+1) and negative correlations are in blue (−1), the intensity of the color and the size of the dot are in accordance with the correlation coefficient. Non-significant correlations (*p* > 0.05) are not shown.

**Table 1 biology-13-00494-t001:** Lipid classes and fatty acid (FA) levels in the liver of asymptomatic fish and those with ‘early’ and ‘advanced’ symptoms of fatty liver disease. Values are means ± SEM. *p*-values that indicate significant differences (*p* < 0.05) are in **bold**. Lowercase letters indicate differences amongst groups. Lipid classes are presented as a percentage of the total lipids. FAs are presented as a percentage of total FAs.

	Asymptomatic	Early	Advanced	*p-*Value
Total Lipids (mg·g^−1^ wet mass)	24.82 ± 4.03	35.00 ± 3.50	38.09 ± 4.70	0.062
Triacylglycerols	17.50 ± 3.50	27.24 ± 5.91	27.74 ± 5.47	0.086
Free FAs	6.56 ± 1.21	6.06 ± 1.27	5.87 ± 0.64	0.878
Sterols	9.50 ± 0.75	6.32 ± 1.12	6.08 ± 0.54	0.066
Acetone Mobile Polar Lipids	6.89 ± 1.04	4.21 ± 1.30	4.07 ± 0.80	0.100
Phospholipids	59.18 ± 4.73	55.72 ± 7.07	55.12 ± 6.12	0.859
Myristic Acid (14:0)	1.75 ± 0.14 ^a^	1.29 ± 0.08 ^b^	1.33 ± 0.09 ^b^	**0.006**
Palmitic Acid (16:0)	20.32 ± 0.70 ^a^	16.65 ± 1.11 ^b^	16.36 ± 0.89 ^b^	**0.013**
Palmitoleic Acid (16:1ꞷ7)	3.41 ± 0.26	3.02 ± 0.19	3.08 ± 0.19	0.374
Stearic Acid (18:0)	5.72 ± 0.25	5.26 ± 0.22	5.33 ± 0.29	0.432
Oleic Acid (18:1ꞷ9)	26.30 ± 1.43	28.73 ± 1.85	31.32 ± 1.44	0.086
Vaccenic Acid (18:1ꞷ7)	3.33 ± 0.14	3.50 ± 0.10	3.59 ± 0.12	0.346
Linoleic Acid (18:2ꞷ6)	7.68 ± 0.42	6.75 ± 0.35	6.94 ± 0.22	0.165
α-Linolenic Acid (18:3ꞷ3)	1.33 ± 0.10	1.21 ± 0.09	1.14 ± 0.07	0.176
γ-Linolenic Acid (18:3ꞷ6)	0.35 ± 0.04 ^b^	0.63 ± 0.12 ^a^	0.78 ± 0.07 ^a^	**0.026**
Eicosenoic Acid (20:1ꞷ9)	1.11 ± 0.12	1.04 ± 0.11	0.87 ± 0.06	0.183
Eicosatetreanoic Acid (20:4ꞷ6)	2.17 ± 0.23	2.58 ± 0.21	2.58 ± 0.20	0.462
Eicosapenteanoic Acid (20:5ꞷ3)	4.31 ± 0.34	4.12 ± 0.28	3.98 ± 0.35	0.339
Docosapentaenoic Acid (22:5ꞷ6)	1.37 ± 0.15	1.54 ± 0.13	1.45 ± 0.16	0.748
Docosahexaenoic Acid (22:6ꞷ3)	13.48 ± 1.49	15.79 ± 0.82	13.75 ± 1.05	0.399
ΣSAT	28.54 ± 0.85 ^a^	23.80 ± 1.29 ^b^	23.61 ± 1.13 ^b^	**0.011**
ΣMUFA	36.61 ± 1.72	38.89 ± 1.83	41.07 ± 1.86	0.242
ΣPUFA	34.44 ± 1.72	36.95 ± 0.94	34.99 ± 1.56	0.549
Σꞷ3	21.66 ± 1.87	24.17 ± 1.03	21.78 ± 1.50	0.480
Σꞷ6	11.99 ± 0.44	11.98 ± 0.20	12.33 ± 0.19	0.575
ꞷ6:ꞷ3	0.65 ± 0.06	0.50 ± 0.03	0.59 ± 0.04	0.462
DHA:EPA	3.06 ± 0.22 ^b^	3.90 ± 0.16 ^a^	3.49 ± 0.15 ^ab^	**0.014**
ARA:DHA	0.17 ± 0.01	0.16 ± 0.01	0.19 ± 0.01	0.227
PUFA:SAT	1.25 ± 0.08	1.58 ± 0.08	1.51 ± 0.09	0.153
Phospholipids:Sterols	6.87 ± 0.89	10.03 ± 1.04	9.34 ± 1.59	0.217
(ΣMUFA + ΣPUFA):ΣSAT	2.54 ± 0.15 ^b^	3.27 ± 0.23 ^a^	3.32 ± 0.22 ^a^	**0.024**

SAT = Saturated FAs, MUFA = Monounsaturated FAs, PUFA = Polyunsaturated FAs, ARA = Eicosatetraenoic Acid (Arachidonic Acid), EPA = Eicosapentaenoic Acid, DHA = Docosahexaenoic Acid.

## Data Availability

Data will be made available on request.

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
