# Peer review of "Prolonged Cold Exposure Negatively Impacts Atlantic Salmon (Salmo salar) Liver Metabolism and Function"

_biology, 2024, doi:10.3390/biology13070494_

Round 1

Reviewer 1 Report

Comments and Suggestions for Authors

The manuscript titled “Prolonged Cold Exposure Negatively Impacts Atlantic Salmon (Salmo salar) Liver Metabolism and Function” described a study focus on Winter Syndrome of Atlantic salmon. The authors carried out a set of gene expression detection and correlation analysis. Unfortunately, I cannot find enough novel insight from this manuscript. And the statistical analysis was also inadequate. Therefore, I think this manuscript cannot be considered for publication in the journal.

Comments on the Quality of English Language

Moderate editing of English language required.

Author Response

We have now responded to the detailed comments provided by the other reviewers, and have further clarified our statistical approaches to ensure both the novelty and statistical rigour of our manuscript.

Reviewer 2 Report

Comments and Suggestions for Authors

 I have gone through the interesting study conducted by team of Rojas et al., which emphases on the prolonged exposure to cold temperatures can lead to the development of FLD in some Atlantic salmon. The study finding and results have been meticulously planned and extensive data set has been generated out of the study. The study holds promise for the development of new diets which might augment the health of salmon during the winter.

I am happy to endorse this manuscript for publication in the journal. However, author may consider some minor changes in the MS.

1. The introduction part is a bit lengthy and needs to be shortened up.

2. The histological changes in the liver architect would have shed more insight to see those changes due to exposure.

3. The basic changes in the liver metabolic enzymes and liver metabolites should be considered for this study.

4. The methodology for  PCoA, PERMANOVA, Pearson correlation matrix  and SIMPER analysis should be included in details

Comments on the Quality of English Language

Minor revision

Author Response

We have carefully addressed all the comments of the reviewer and have revised the manuscript accordingly. We believe that the quality of the manuscript has improved considerably in responding to the reviewers’ comments and requests for additional information. Our responses to the reviewers’ comments/questions/suggestions are provided in a point by-point format in attached file.

Reviewer 3 Report

Comments and Suggestions for Authors

This study involves examination of 34 genes and lipid samples from a study previously published by this group in the journal Aquaculture.  The previous study had the objective of experimentally lowering water temperatures to 1oC and examining physiological changes, stress and mortality.  They reported some gene expression and concluded serpinh1 and cirbp were not good biomarkers for decreasing temperature and indicated the symptoms observed in the fish were “suggestive of liver dysfunction”.   This current manuscript examines 34 additional genes in search of biomarkers to indicate liver dysfunction at cold temperatures. It also correlates some liver / lipid indices with gene expression.  In this respect, the data presented are worthwhile.  However, other aspects of the manuscript are problematic. The PIs try to tie this study to the occurrence of winter syndrome or winter disease.  There is no mention that the symptoms they scored in the fish used in this study (and the previous one) are consistent with winter syndrome. Is this the same disease? The fish used are from New Brunswick. Are these fish genetically similar or the same stocks of fish exhibiting winter syndrome in Eastern Canada and Iceland?  Fish used in this study were binned into three groups based on external symptoms and behavior then physiological indices (lipids) and expression compared between bins.  The fact that 27 fish were equally binned into three groups suggests that selection of a subsample of fish was not random. Moreover, scoring is arbitrary, categorical and qualitative. Why not run all fish together then a posterori, determine if they fall out into bins of asymptomatic, early and late?  How do you know an asymptomatic is asymptomatic just based on behavior and general appearance? 

The previous paper this study borrows heavily from and cited numerous times is titled in the references (#41) “Low Seawater Temperatures are associated with liver lipid accumulation and dysfunction, ionoregulatory disturbance and opportunistic infections in cultured post-smolt Atlantic salmon.” The reference is an incomplete citation because it does not exist on any of my search engines. Thus, looking up where the stock of Atlantic salmon originated is difficult unless you read: Lowering temperature to 1 °C results in physiological changes, stress and mortality in cultured Atlantic Salmon (Salmo salar) which I think is the paper they wanted to reference. It could also be from the dissertation on file at Memorial University. 

In all, this is not a standalone study. Moreover, any link to winter syndrome in Icelandic fish and these fish remains unconvincing. I’m also not sure how the fish for this study were chosen as a subgroup from the previous study other than by observation of fish showing signs of ill health after five weeks at 3oC.

Author Response

(The authors gave the same response as above.)

Reviewer 4 Report

Comments and Suggestions for Authors

The manuscript by Rojas et al., describes a study investigating whether Atlantic Salmon can develop fatty liver disease following prolonged cold-exposure as a potential mechanism to explain mortalities associated with such exposures.  The authors measured different classes of hepatic lipids, fatty acid levels and the mRNA expression of 34 molecular markers associated with fatty liver disease.  They then correlated their results with disease associated characteristics identified from previous studies.

This is an interesting and timely study that advances our knowledge of the effects of prolonged cold exposure on an important fishery, Atlantic Salmon.  I found no issues with the experimental design, methodology or interpretations of results or conclusions.  Overall, this is a well-written and well-organized manuscript that was a pleasure to read.  I particularly appreciate the thorough description of the qPCR methods, something that is too often lacking.  I only have a few very minor points for the authors to consider below. 

Line 142:  I assume evenly distributed (75/tank)?

Section 2.4.2:  what type of primers were used – random hexamers?

Line 278:  were amplicons sequence verified?

Lines 919-923:  I guess one thing that’s not clear to me is how these biomarkers would be used – just as a tool for future studies or is there a practical application for aquaculture management that I’m missing?

Author Response

(The authors gave the same response as above.)

Round 2

Reviewer 3 Report

Comments and Suggestions for Authors

Revisions to this manuscript have been well executed.  I have only only minor edit on Line 348: "All the data were..."   Data are always treated as plural.